# Quantifying the analysis uncertainty for nowcasting application

Yanwei Zhu[1,2], Aitor Atencia[3], Markus Dabernig[3], Yong Wang[1,4]

[1]School of Atmospheric Science, Nanjing University of Information Science and Technology, Nanjing, China
[2]HuaFeng Research Lab for Weather Science and Applications, Nanjing University of Information Science and Technology, Nanjing, China
[3]GeoSphere Austria, Vienna, Austria
[4]CMA Earth System Modelling and Prediction Centre, China Meteorological Administration, Beijing, China

*Correspondence to*: Yong Wang (yong.wang@nuist.edu.cn)

**Abstract.** This study proposes a method to quantify uncertainty represented by errors in very high-resolution near-surface analysis, specifically for weather nowcasting applications. Gaussian-distributed perturbations are used to perturb the first guess and observation, with variance equal to that of the first guess error. This error not only reflects the spatial characteristics of the difference between first guess and observation but also dominates the primary uncertainty in analysis errors. However, mapping perturbations to analysis grid mesh through interpolation results in under-dispersion, particularly in areas without stations. To address this issue, Gaussian perturbations are inflated with an inflation factor to amplify the dispersion. This method was applied to high-resolution analysis and nowcasting for hourly temperature, humidity and wind components in the Beijing-Tianjin-Hebei region, to assess its effectiveness in representing uncertainty. The generated ensemble analysis exhibits reasonable spread and high reliability, indicating the accurate quantification of analysis uncertainty. Ensemble nowcasting is extrapolated from ensemble analysis to evaluate the transmission of perturbation during extrapolation. Verification results of ensemble nowcasting reflect that the spread increases effectively during extrapolation up to a lead time of 6 hours. However, the increase of spread is highly dependent on the persistence of numerical weather prediction. The results demonstrate that generating appropriate perturbations based on analysis errors effectively represents the analysis uncertainty and contributes to estimating uncertainty in nowcasting.

## 1 Introduction

Nowcasting is crucial for severe weather warnings and for protecting life and property, as it rapidly predicts high–impact weather events in near-real time (Wang et al., 2017a, b; Wastl et al., 2018; Schmid et al., 2019). A very high–resolution weather analysis forms the basis for skilful nowcasting, providing accurate real–time atmospheric conditions at the initial time (Wastl et al., 2021). However, due to the chaotic nature of the atmosphere, errors in data, and imperfect numerical models, nowcasting involves uncertainties (Lorenz, 1965; Leith, 1974; Kann et al., 2012; Glahn and Im, 2013; Wastl et al., 2019). To deal with uncertainties, generating an ensemble using appropriate perturbations is an effective method (Leutbecher et al., 2007; Leutbecher and Palmer, 2008).

In recent years, the use of ensemble nowcasting has become increasingly widespread (Wang et al., 2017b, 2021; Yang et al., 2023). Numerous studies have demonstrated that addressing all sources of the uncertainty represented by errors is a key aspect for generating ensemble (Sun et al., 2014; Thiruvengadam, 2020). The weather analysis contains uncertainty, which significantly impacts nowcasting due to both measurement errors and computational errors (Eibl and Steinacker, 2017; Keresturi et al., 2019). As a result, quantifying uncertainty caused by these errors in analysis is one of the major challenges in constructing ensemble nowcasting (Wang et al., 2017a; Taylor et al., 2022).

A widely applied approach for quantifying uncertainty is introducing appropriate perturbations to generate ensemble, based on the characteristics of errors (Buizza et al., 2005; Zhu, 2005; Bouttier et al., 2016; Chen et al., 2016; Wang et al., 2017a; Lin et al., 2022). Most studies focus on addressing uncertainty in nowcasting, but few have explored the impact of analysis errors (Bouttier, 2019; Wang et al., 2021). Wang et al. (2014) and Suklitsch et al. (2015) presented evidence that introducing additional perturbations to estimate the analysis uncertainty can improve the simulation of nowcasting uncertainty. Saetra et al. (2004) explored the influence of observation errors on analysis uncertainty. Horányi et al. (2011) and Bellus et al. (2016, 2019) demonstrated that appropriate perturbations can simulate observation uncertainty in analysis. The Aire Limitée Adaptation dynamique Développement InterNational-Limited Area Ensemble Forecasting (ALADIN-LAEF) and Convection-permitting Limited-Area Ensemble Forecasting system (C-LAEF) are two skilful systems that use 16 members to represent analysis uncertainty. These systems account for uncertainty by perturbing observations, while the 16 first guesses provide important uncertainty information within the three-dimensional background (more details in Wang et al., 2011; Bellus et al., 2016; Wastl et al., 2021). However, neither ALADIN-LAEF nor C-LAEF addresses the impact of other sources of uncertainty, such as those arising from interpolation (Wastl et al., 2021). Therefore, considering various types of analysis errors is crucial for more accurately quantifying analysis uncertainty (Suklitsch et al., 2015).

An accurate analysis, which describes the current atmosphere, is typically derived by assimilating the first guess and observation data (Randriamampianina and Storto, 2008; Kann et al., 2009; Lin et al., 2022). The observations provide an estimate of the true atmospheric values, while the three-dimensional first guess provides a complete spatial structure within the region of interest (Sun et al., 2013; Hoteit et al., 2015; Casellas et al., 2021). However, when combining observations with terrain-corrected first guesses, interpolation errors caused by the algorithm may also arise (Leutbecher and Palmer, 2008; Feng et al., 2020). Current research has not clearly addressed the impact of interpolation error uncertainty on the analysis and nowcasting. Hence, it is crucial to investigate how to accurately estimate interpolation error in analysis to gain a comprehensive understanding of both analysis and nowcasting uncertainty.

Integrated Nowcasting through Comprehensive Analysis (INCA) system calibrates the first guess using automatic weather station observations (Haiden et al., 2010, 2011). The first guess in INCA is the numerical weather prediction (NWP) field, which is provided by the Austrian operational version of the ALADIN limited-area model, as described by Wang et al. (2006). In the NWP calibration, a topographic characteristic factor is used to correct the surface layer to match the actual terrain (Kann et al., 2009). Seamless Integrated Weather Prediction and Applications (SIVA) is a multivariable analysis and nowcasting system based on INCA framework, and it is applied in the Beijing-Tianjin-Hebei region, China (BTH). The NWP output of

China Meteorological Administration Mesoscale model (CMA-MESO) provides a deterministic first guess, which is used by SIVA to describe the spatial characteristics (Shen et al., 2020). Since the first guess is deterministic, the uncertainty of SIVA analysis is not considered. Therefore, it is necessary to consider the computational errors to accurately quantify the uncertainty in SIVA analysis.

The current methods for estimating uncertainty in analysis primarily depend on perturbations either derived from the first guess or generated based on the inherent errors in observation data (Horányi et al., 2011; Wang et al., 2017a; Wastl et al., 2018; Yang et al., 2023). These approaches introduce perturbations without accounting for the uncertainty intrinsic to the analysis calculation process. Nevertheless, they provide valuable insights, suggesting that perturbations can be generated according to the statistical characteristics of errors. Building upon this foundation, this study proposes a novel approach that generates

perturbations based on errors in the calculation process itself, offering a more comprehensive way to quantify analysis uncertainty. This approach not only addresses the limitations of current techniques but also enhances the precision of uncertainty representation, with significant potential for improving ensemble nowcasting applications.

This article is organized as follows. Section 2 introduces the algorithm of SIVA analysis. Section 3 elaborates on the characteristics of errors and perturbation methods. Section 4 is dedicated to the verification results of ensemble analysis and

nowcasting. A summary and conclusions are given in section 5.

## 2 Method and data

SIVA provides hourly analysis and nowcasting fields at a horizontal resolution of 1 km for near-surface temperature, humidity and wind speed components. The analysis starts with a first guess, which is an NWP short-range forecast output of CMA-MESO. The CMA-MESO runs twice daily with a forecast range of 48 hours, starting at 00:00 and 12:00 (UTC). It has a

horizontal resolution of 3 km and 51 vertical levels. Forecast field of CMA-MESO is interpolated to SIVA grid mesh at a horizontal resolution of 1 km to serve as the first guess, which is then calibrated based on its errors relative to observations. Topographic parameters are used to map the height of CMA-MESO model levels to the actual altitude of the station location. The observations used in this research, considered as ground truth, are provided by automatic weather stations and include hourly 2-meter (2m) surface temperature, specific humidity, and 10-meter (10m) surface wind speed. These data can be

obtained by submitting an application through the official data platform of the China Meteorological Administration. The algorithm for the 2m temperature, humidity, and 10m wind speed analysis module of SIVA consists of the following steps:

Step 1: Revise the first guess by inverse distance weight interpolation (IDW) using ground observation data to obtain a three-dimensional revised field (3DRF). The interpolation process in this step is referred to as 3D interpolation.

Step 2: Combining the topographic features and observation data, the 3DRF is revised again and then interpolated

vertically to the lowest model level to obtain the near–surface revised field (NRF). The interpolation process in this step is denoted as 2D interpolation.

Since SIVA is a new version of INCA developed in BTH region, readers can find more details of the algorithm in Haiden et al. (2010, 2011). To avoid the confusion of the term "error", the difference between first guess and observation in Step 1 is denoted as "error1". The difference between 3DRF and observation in Step 2 is denoted as "error2". The experimental periods are August 2020 (hereafter called summer) and February 2021 (hereafter called winter). With the high horizontal resolution, SIVA can describe the geographic features in detail. In addition, 21 vertical levels corresponding to various altitudes, such as 0 m, 200 m, and up to 4000 m above the ground, are used to match stations at different elevations. The wind speed is represented in 32 vertical levels with intervals of 125 m. This approach ensures that the first guess is calibrated using topographic parameters and observation at station location. This study discusses three meteorological elements: temperature (T2m), humidity (relative humidity RH2m and specific humidity QQ2m) at 2 meters above the surface, and the wind speed components (U10m and V10m) at 10 meters above the ground.

BTH has a unique and complex topographic structure, with the northwestern part between two mountain ranges and the southeastern part belonging to the North China Plain (Fig. 1). Such topographic information and station altitude are used to impose terrain constraints on the SIVA grid points. There are 1670 automatic weather stations can pass quality control and are used for T2m and humidity analysis. For wind components, 2351 stations are available. To assess the effectiveness of ensemble analysis in representing uncertainty, 151 stations are randomly selected as the test set, while the remaining 1519 stations are the training set and then are used to generate the ensemble analysis and nowcasting (Fig. 1). Since wind components have different quality control framework with temperature, the total stations for wind are 2350 while the number of test stations is 191.

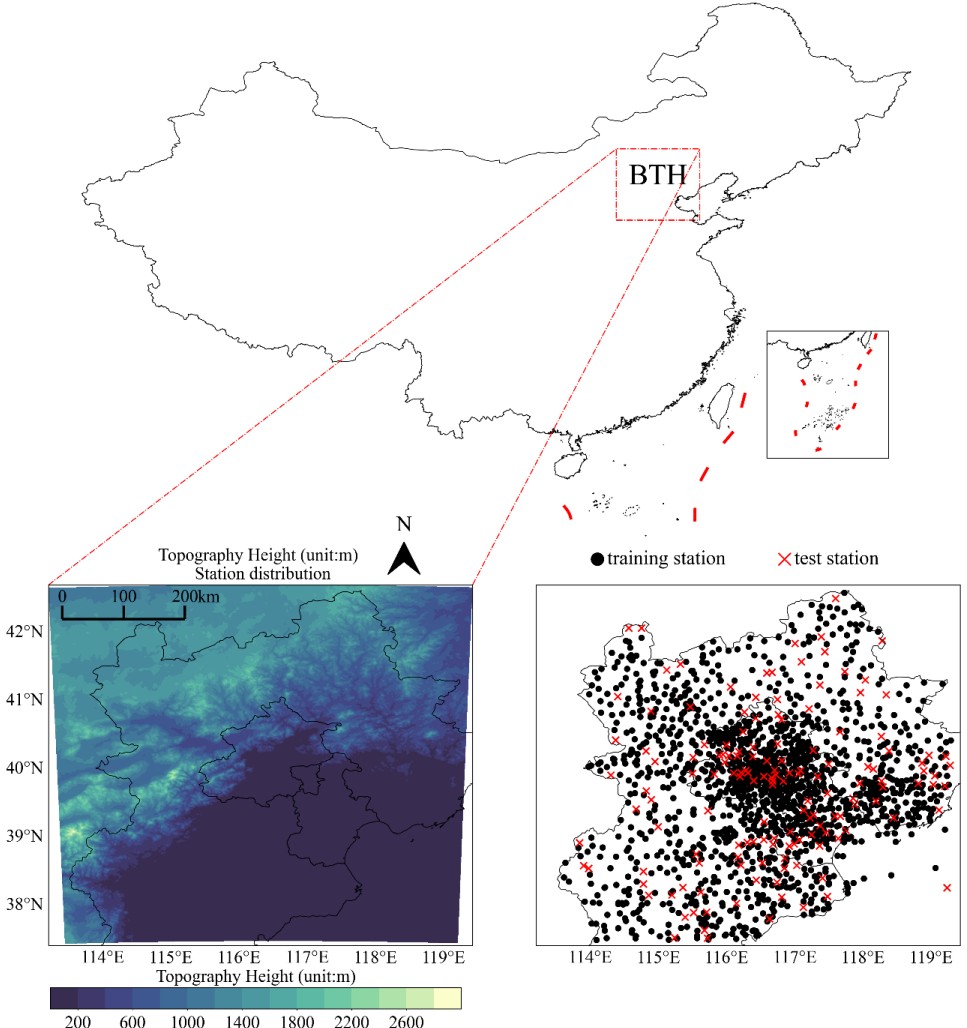

**Fig. 1. Topography height (unit: m) and station distribution over the Beijing-Tianjin-Hebei (BTH) region. Black dots represent training stations and red dots represent test stations, both used for T2m and QQ2m.**

This work proposes a perturbation method to accurately quantify the uncertainty represented by the errors in analysis. These errors depend on the observations, which can only be obtained in areas with station information and interpolated across the entire grid mesh. Therefore, the magnitude of perturbation is expected to be consistent with the statistical characteristics of these errors, as observed at the training stations (Fig. S1). In addition, the test stations assess whether interpolation can effectively propagate the perturbation information throughout the entire space. To evaluate the impact of the perturbation in

extrapolation, the perturbed analysis is used to generate ensemble nowcasting. The ensemble nowcasting starts hourly and extrapolates up to a lead time of 6 hours. The verification of both ensemble analysis and nowcasting covers the test stations shown in Fig. 1.

**3 Perturbation method**

For T2m, error1 is based on the principle of minimal required correction to filter out the excessive forecast errors, which may
result in some instances where the error value is zero, even though the true error1 value is nonzero. Therefore, the forecast error of NWP causes a major source of uncertainty in error1. Due to the terrain constraint, the spatial distribution of error1 and error2 exhibits distinct topographic features, and the analysis error will reflect these geographic characteristics (Fig. S1). As described by Horányi (2011), perturbations are generated by the standard deviation (STD) of errors to represent the uncertainty at observation site. Hence, the observation perturbation in this study is generated based on the STD of error1 at the training
station sites. Error1 is the difference between first guess and observation, and is represented at training station locations. This means that at each training station the observation perturbation is sampled randomly from Gaussian noise with a mean of 0 and a scale equal to the STD of error1. However, both the error1 and observation perturbation are extrapolated to the entire grid mesh through interpolation, which means that the areas without stations (hereafter called area with test stations) can only have partial information of the error1 and perturbation. Therefore, to account for uncertainty across entire grid mesh more
comprehensively, the perturbation of first guess is crucial, as the first guess is responsible for providing a calibrated forecast in areas without observations.

To estimate the uncertainty across the entire space, the STD of error1 is interpolated into the SIVA grid mesh to perturb the first guess. At each grid point, the perturbation (hereafter called first guess perturbation) is a Gaussian noise with a mean of 0 and STD equal to that of the interpolated error1. Since error1 is the distance between first guess and observation at the lowest
model level, it implies the downward (upward) shift that aligns model height to true altitude of the station location. This indicates that in vertical interpolation the forecast error will combine with geographic feature. In addition, interpolating 3DRF to surface level depends on the vertical errors between adjacent levels. Hence, the first guess perturbation is sampled from Gaussian noise with a shape of (M, Z), where M represents the number of members, and Z denotes the number of levels. Such first guess perturbation can represent the NWP systemic uncertainty in vertical dimension, while the horizontal distribution of
interpolated error1 provides the spatial characteristic.

As described above, both observation perturbation and first guess perturbation are generated based on the errors in analysis, and within areas that have observation information. Introducing these perturbations into the observation and first guess, ensemble analysis is constructed. To evaluate the representation of the uncertainty, the ensemble is interpolated to all stations, and used the observation data to verify the reliability of it. Figure 2 shows the root-mean-square error (RMSE) and ensemble
spread in both training and test stations. The spread quantifies the dispersion or variability among the ensemble members, while the RMSE represents the errors of the ensemble mean. Comparing spread and RMSE assesses the statistic reliability of the ensemble. A reliable ensemble should have exhibit alignment between the spread and the RMSE (Fortin et al., 2014). It can be observed that the spread in training stations is consistent with the RMSE (Fig. 2a, b). The ratio of RMSE and spread is nearly equal to 1 (Fig. 2c). It indicates that the uncertainty in area with station information has been appropriate quantify.
There is a significant under-dispersion in test stations (Fig. 2d, e). This under-dispersion can be interpreted as the information

in the area without stations is not sufficient. But in the concept of appropriate represent the uncertainty, the spread in test stations should be amplified to be equal to the RMSE.

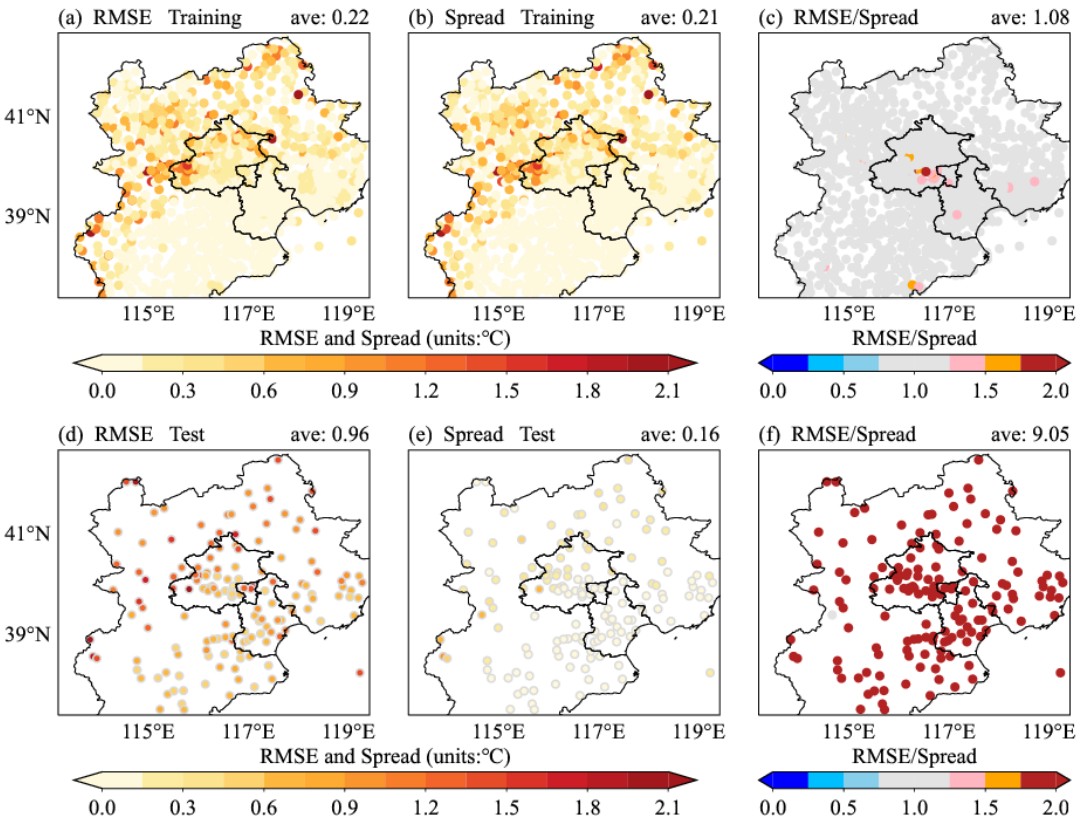

Fig. 2. RMSE of the ensemble mean (a, d), spread of the ensemble (b, e) and the ratio of RMSE to spread (c, f), for T2m at both training and test stations. All values are averaged over August 2020.

An inflation factor is calculated to address the under-dispersion at test stations. The inflation factor is the ratio of RMSE to spread ($R_{rs}$) at test stations in the ensemble analysis obtained by perturbing both first guess and observation (e.g., as shown in Fig. 2f). This factor is then interpolated to the grid mesh to amplify the STD of first guess perturbation in area with test stations, thereby ensuring that the spread aligns with RMSE (Fig S2).

By combining inflation factor with the perturbations, they are generated as follows:

$$\mathbf{NWP}'_{(i,j)} = \mathbf{NWP}_{(i,j)} + \mathbf{N}(0, STD_{2D-error1(i,j)} * infla\_factor_{(i,j)}), \tag{1}$$

$$\mathbf{OBS}'_k = \mathbf{OBS}_k + \mathbf{N}(0, STD_{error1(k)}). \tag{2}$$

Here, 2D–error1 represents the interpolated STD of error1. The subscript $(i, j)$ and $k$ refer to the SIVA grid point and the $k$th station, respectively. It should be emphasized that the observation of wind has not been perturbed. The true values of wind

components are obtained through the sine and cosine transformations. Therefore, perturbing the wind components by error–based noise cannot reflect the uncertainty arising from wind direction.

     The principle of minimal required correction in T2m analysis results in a significant reduction in the dispersion of perturbed error1 relative to unperturbed error1. Nevertheless, the impact of the inflation factor guarantees that the dispersion of ensemble errors will not attenuate excessively (Fig. 3a, b). The error dispersion at both training and test stations aligns with the perturbed
error1. This consistency substantiates the reliability of this approach involving the introduction of the inflation factor in representing uncertainties. The generated ensembles consistently represent uncertainty with comparable reliability across the entire region (Fig. 3). Furthermore, the results of ensembles with varying members differ by only about 1%. To balance computational efficiency with the need for sufficient members, the ensemble size is set 20.

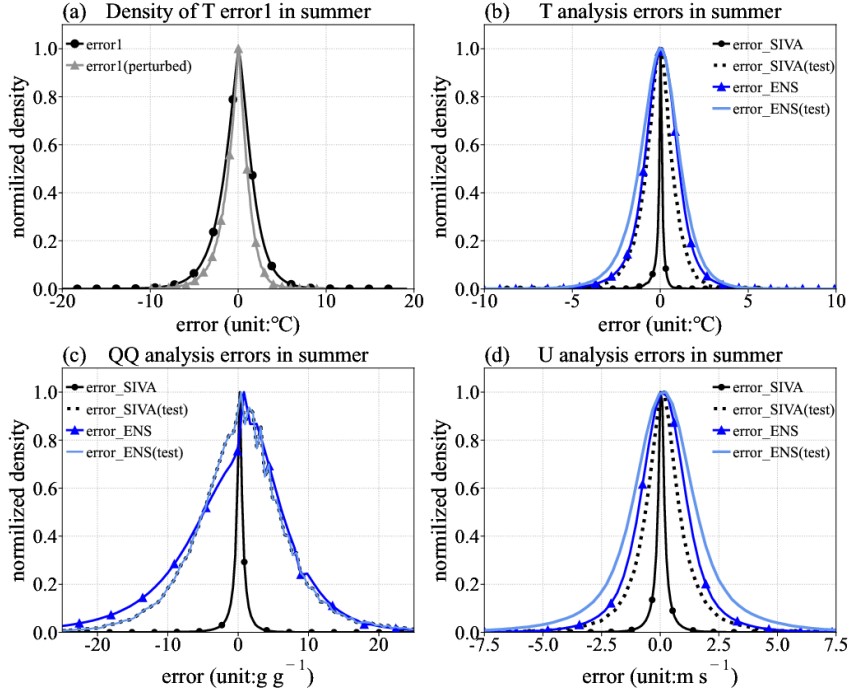


**Fig. 3. Probability density function (PDF) of errors. (a) shows the error1, where the thick black line (●) represents the unperturbed error1, and the thin grey line (▲) refers to the perturbed error1.  (b) ~ (d) show the analysis errors of temperature, specific humidity, and the U component of wind speed, respectively. The black solid and dashed lines represent the PDFs of SIVA deterministic analysis errors at both training and test stations, respectively. The blue and light blue lines represent the PDFs of the ensemble analysis**
**errors at the same stations.**

## 4 Verification

### 4.1 Verification of ensemble analysis

The deterministic SIVA analysis (hereafter called CANA) served as a reference for assessing the performance of the generated ensemble in estimating uncertainty at test stations. Commonly used probability verification scores, including RMSE, ensemble spread, Talagrand diagram or rank histogram, and reliability diagram, are applied to evaluate the effectiveness of uncertainty representation at test stations (Hamill et al., 2000; Hamill 2001; Fortin et al., 2014).

### 4.1.1 Ensemble RMSE and Spread

For the verification of T2m, QQ2m and RH2m ensembles, a total of 151 stations are used. One station in southeast is only available in summer and there are 150 test stations in winter. There are 191 test stations to evaluate the performance of wind components. Table 1 presents the averaged RMSE and ensemble spread for the meteorological elements discussed in this research.

Table 1. RMSE and spread of the ensemble analyses for summer (August 2020) and winter (February 2021). The subscript C (E) denotes the RMSE of CANA (ensemble analysis), and $R_{rs}$ represents the ratio of ensemble RMSE to spread. Values are averaged over all the test stations and the entire period.

| | Summer | | | | Winter | | | |
|---|---|---|---|---|---|---|---|---|
| | $RMSE_C$ | $RMSE_E$ | Spread | $R_{rs}$ | $RMSE_C$ | $RMSE_E$ | Spread | $R_{rs}$ |
| T2m (°C) | 0.96 | 0.96 | 0.89 | 1.06 | 1.23 | 1.23 | 1.18 | 1.03 |
| RH2m (%) | 6.92 | 6.73 | 4.58 | 1.37 | 6.57 | 6.57 | 5.67 | 1.17 |
| QQ2m (g g$^{-1}$) | 1.09 | 1.08 | 0.95 | 1.12 | 0.24 | 0.23 | 0.23 | 0.99 |
| U10m (m s$^{-1}$) | 1.25 | 1.28 | 1.29 | 0.99 | 1.90 | 1.96 | 1.96 | 0.99 |
| V10m (m s$^{-1}$) | 1.37 | 1.41 | 1.40 | 0.99 | 1.85 | 1.91 | 1.90 | 1.00 |

Without inflation, the averaged ensemble spread is around 0.16, while the RMSE is approximately ten times larger (Fig. 2 d ~ f). This reflects the considerable under-dispersion of the ensemble (Fig. 4a, d). By introducing the inflation factor, the ensemble spread increases to 0.89 and $R_{rs}$ approaches 1, indicating that the ensemble exhibits reliable dispersion. Moreover, the increase in spread of both temperature and humidity does not result in a corresponding increase in RMSE (Table 1). This

is attributed to the zero mean of the perturbations, allowing the inflation factor to adjust only the STD without affecting the mean.

The primary objective of this work is not to show dramatic improvements in verification scores, but to focus on the quantification of uncertainty in analysis. With the effect of inflation factor, the perturbation in the corresponding grids of the test stations is amplified to align the ensemble spread with RMSE. In addition, the perturbed first guess reflects the spatial uncertainty at the grid points. Hence, the distance weight in 3D interpolation of 3DRF can account for the uncertainty derived from the geographic characteristics of CMA-MESO. Most test stations have spread nearly equal to the RMSE, indicating the accuracy of the uncertainty quantification (Fig. S3). In mountainous areas, the RMSE is negatively affected due to the lack of precise information about inversion heights (Fig. S3 g). A marine station in the southeast receives perturbation information mainly from 3DRF and only a small amount of NRF information during the summer.

The QQ2m exhibits similar characteristics to T2m, but limitations in variable conversion affect the transmission of perturbation information, resulting in underestimated dispersion for RH2m. Nevertheless, the ensemble of humidity maintains the errors consistency with CANA and provides a reliable estimate of uncertainty (Fig. S4).

For wind components, uncertainty is represented by the perturbation of first guess. The consistency of ensemble spread and RMSE indicates the effective representation of the error uncertainty. But the increased RMSE of ensemble mean may result from the lack of observation perturbation or the divergence constraint of wind components (Table 1). One possible approach is to account for the error arising from the wind components conversion to support the generation of observation perturbation.

### 4.1.2 Probability scores

The Talagrand diagram describes the characteristics of ensemble spread and bias (Hamill 2001). It evaluates the ability of an ensemble to reflect the observed frequency distribution. A flat rank histogram is the indication of a perfect ensemble, with the uniform reference rank is equal to $1/(M+1)$, where M is the ensemble size. In this study, the uniform rank is 0.0476, which corresponds to the dashed line in the Talagrand diagram. The "U-shaped" rank histogram illustrates that the ensemble without inflation is under-dispersive and does not sufficiently represent the uncertainty, consistent with the results shown in Fig. 2 (Fig. 4a, d). In contrast, the inflated ensemble presents a nearly flat rank histogram (Fig. 4b, e). This result indicates that the ensemble exhibits reliable dispersion, which aligns with the scores presented in Table 1.

The reliability diagram illustrates how the ensemble probabilities match the frequency of verification reference at a given threshold. For a perfect ensemble, probabilities should equal the verification frequency (the diagonal line in the diagram). The samples within each bin, as shown in the sharpness histogram, represent the resolution of the ensemble. The thresholds used in the diagram correspond to the median value of the considered elements. Figure 4c and 4f show that the ensemble with inflation exhibits high reliability, whereas the ensemble without inflation displays poor reliability and resolution. Both the Talagrand and reliability diagrams demonstrate that the inflation factor effectively amplifies the spread at test stations, thereby enhancing the capability of ensemble analysis to represent uncertainty in areas lacking station information.

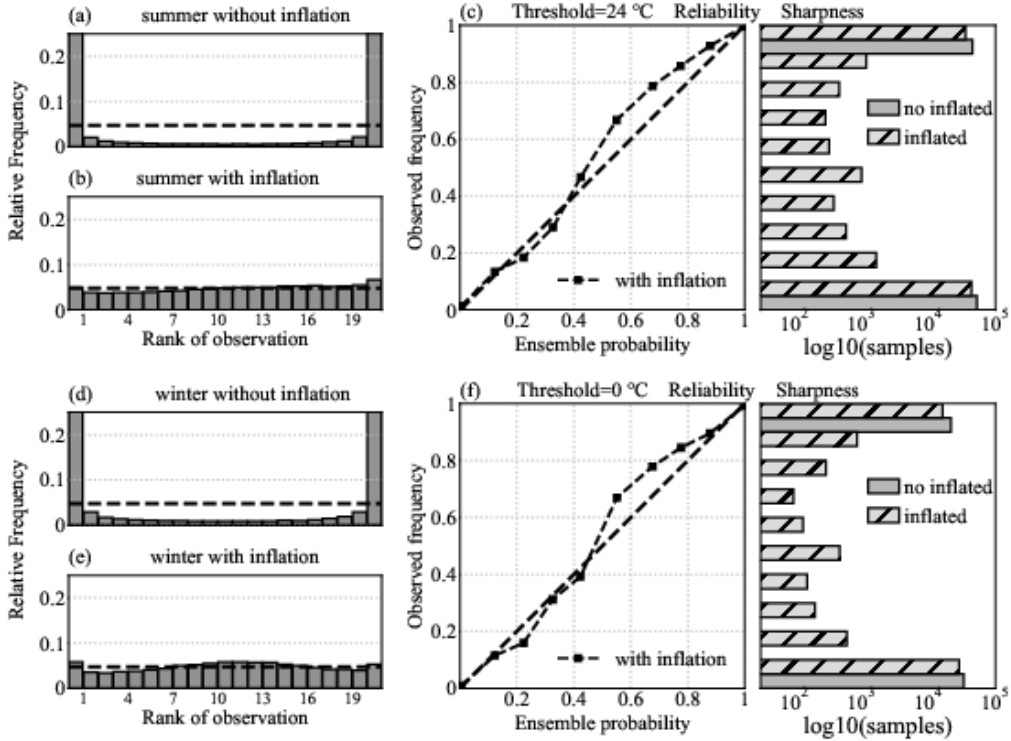

Fig. 4. Talagrand (a, b, d, e) and reliability diagrams (c, f) for T2m ensemble analysis, averaged over all test stations.

As stated above, the inflation factor is based on test stations and is extrapolated to entire grid mesh via interpolation. To gain a comprehensive understanding of the approach proposed in this study, it is important to evaluate ensemble performance in areas where neither training nor test stations are available. An experiment is conducted by dividing the stations into three groups: 1) training stations that participate in the computation of analysis; 2) test stations used for calculating the inflation factor; 3) outside stations that are excluded from both the analysis and the inflation factor calculation. The outside stations are actually representing the areas with no observation information, where the data in these areas generally tend to align with the first guess. Verifying the ensemble at outside stations evaluates the representation of uncertainty in areas dependent solely on interpolation. Figure 5 shows the rank histogram for T2m at outside stations during summer, both without and with the inflation factor. The rank histogram illustrates that the inflation factor increases the dispersion at outside stations. However, the spread is around 0.73, less than that of for test stations (0.89). In addition, as observed in Fig. 5b, the ensemble is slightly under-dispersive and exhibits a cold bias. The results highlight a limitation of the approach: the inflation factor relies on stations not used in the analysis computation but exist in practice. This factor cannot adequately represent the complexity of interpolation uncertainty in areas where there are truly no stations. A potential improvement would be to explore whether there are some predictors could help inflation factors to extrapolate this information to the outside stations site.

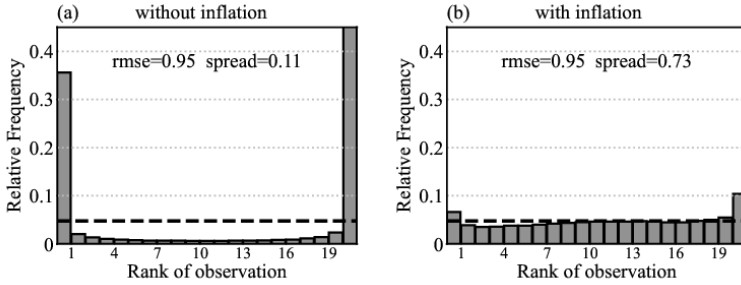

**Fig. 5. Talagrand diagram for T2m ensemble analysis without (a) and with (b) inflation in summer, averaged over all outside stations.**

Figure 6 shows the Talagrand diagram and reliability diagram for QQ2m and RH2m, in both summer and winter. The ensemble for QQ2m in summer has high resolution and reliability at different thresholds. Additionally, the flatness of the rank histogram indicates that the uncertainty has been estimated accurately. The conversion of RH2m involves temperature. To avoid the influence of temperature uncertainty on the variable conversion, only deterministic temperature observation is used in the humidity module. Therefore, this study does not account for the interplay effects among different variables. As a result, systematic biases in T2m within SIVA will propagate into RH2m, causing a dry bias in winter. In addition, the rank histogram and reliability for RH2m illustrate that the ensemble is under-dispersive, due to the neglect of variable conversion.

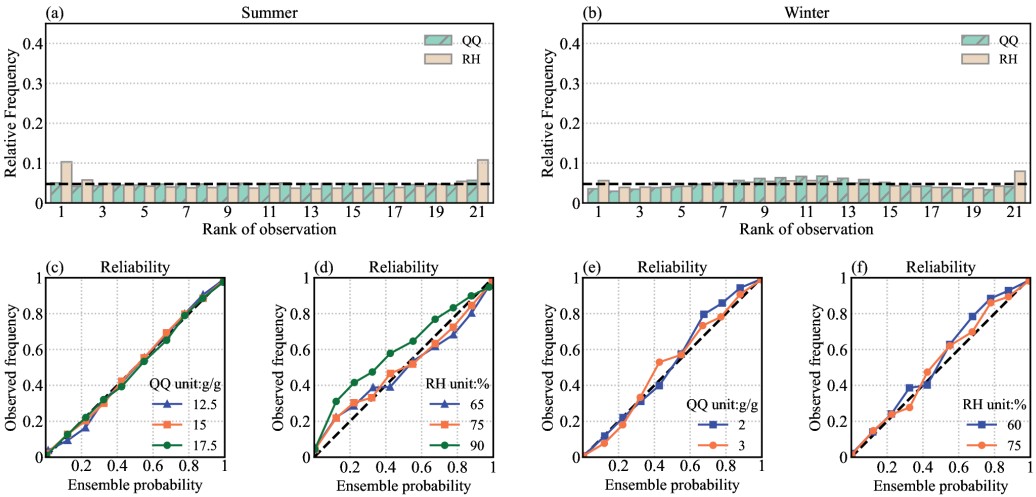

**Fig. 6. Talagrand (a, b) and reliability diagrams (c ~ f) for the ensemble analysis of QQ (unit: g g⁻¹) and RH (unit: %), averaged over all the test stations.**

Figure 7 shows the Talagrand diagram and reliability diagram for wind components in summer and winter. A positive value of U10m (V10m) represents the westerly (southerly) wind, while a negative value means easterly (northerly) wind. The bias for wind ensembles observed in the reliability curve is primarily caused by the representation of uncertainty in wind direction.

For example, when U10m is +1 m s⁻¹, the ensemble values may show -1 m s⁻¹, leading to significant bias due to the opposing wind direction. Although both U10 and V10 display a flat rank histogram, the $R_{rs}$ is less than 1 (Table 1) and reliability curve does not match to the diagonal line. These results suggest that the ensemble analysis of wind components is less effective in reflecting uncertainty compared to T2m. A probably resolution is to consider the impact of wind direction on observation data.

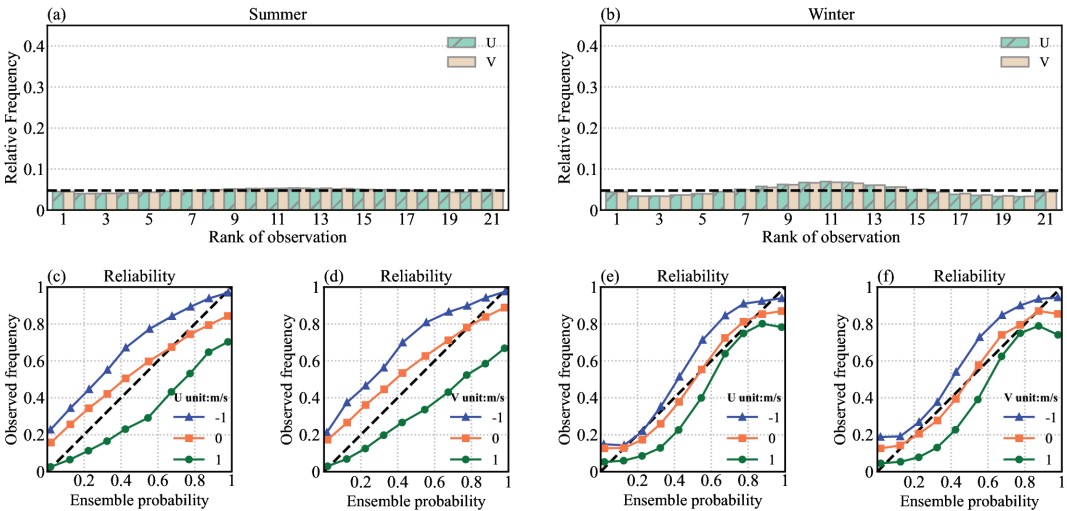

**Fig. 7. The same as Fig. 6 but for the U10m and V10m (unit: m s-1).**

## 4.2 Verification of nowcasting

Verifications of ensemble analysis demonstrate the reliability of the uncertainty estimation. The ensemble provides a spread
at the initial time of nowcasting and exhibit a reasonable error that is consistent with the deterministic analysis. To evaluate the transmission effect of the spread in forecast extrapolation, the perturbed analyses are employed to compute ensemble nowcasting. Due to the range limitation in variable conversion, direct ensemble nowcasting for RH2m is infeasible. For wind, SIVA assigns a weight to the first guess based on an extrapolation step function. As the effectiveness of the extrapolation method gradually diminishes with the prolongation of time steps, the weight assigned to uncertainty information decrease
when extrapolating to a lead time of 3 hours. Hence, the ensemble nowcasting of wind is limited to a lead time of 2 hours, while T2m and QQ2m are extrapolated up to a lead time of 6 hours.

### 4.2.1 Ensemble BIAS and RMSE

Figure 8 shows that the bias and RMSE of ensemble mean are nearly identical to those of deterministic nowcasting (hereafter called CNOW). As described at Section 4.1, the primary objective of this work is to quantify the uncertainty using a
300 perturbation approach. The introduced perturbations are Gaussian–distributed with a mean of zero. In this context, the scores

(BIAS and RMSE) of the ensemble mean should remain consistent with those of the deterministic reference. This ensures that the perturbations do not introduce additional biases, while maintaining an accurate representation of uncertainty.

The increases in ensemble spread for both training and test stations are consistent with those of the deterministic references and RMSE (Fig. 8). However, a certain degree of under-dispersion can be observed as the lead times increase. One reason is that the extrapolation is based on the persistence of first guess, which can only provide deterministic information. Although first guess is perturbed at the initial time of the nowcasting, no additional noise is introduced in the forecasting. Therefore, the nowcasting uncertainty totally depend on the ensemble analysis. Furthermore, the perturbations in areas with test stations are amplified by inflation factor. However, in test stations, the difference between spread and RMSE is more pronounced than in training stations. This phenomenon indicates that, although the inflation factor can amplify the spread in areas without stations, it does not fully capture the uncertainty from the complex interpolation in the analysis calculation. The ensemble nowcasting for specific humidity are also calculated, and due to its similarity to the temperature calculation framework, the results are generally consistent.

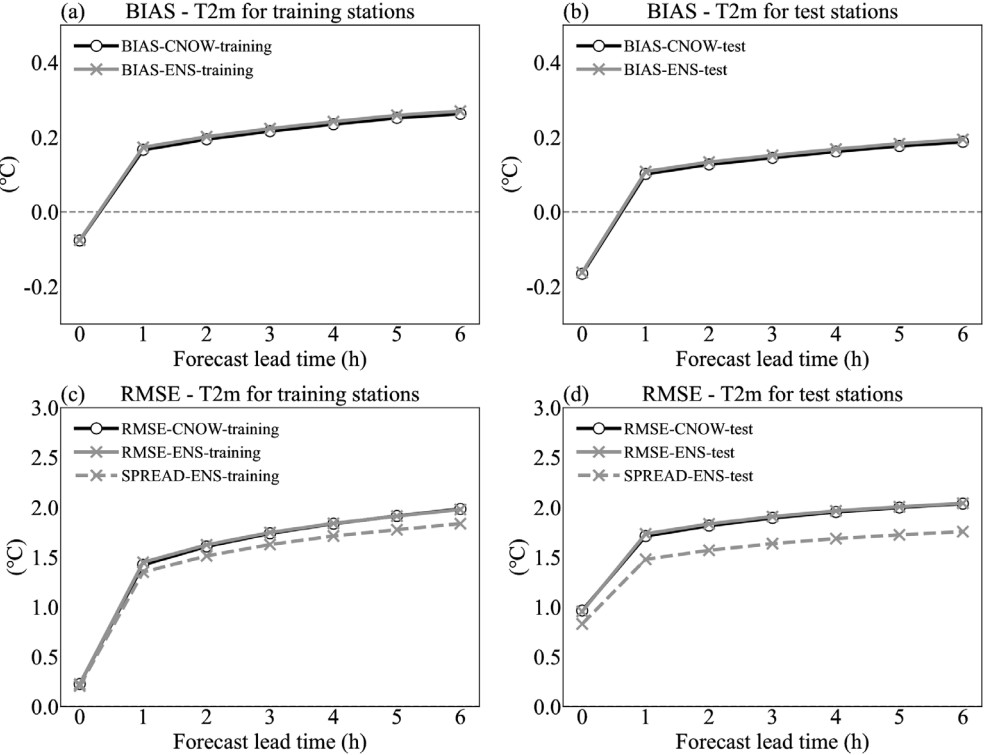

Fig. 8. BIAS (a, b) and RMSE (c, d) for the deterministic reference (black, CNOW, ○) and ensemble mean (grey, ENS, ×) as a function of forecast lead time. The ensemble spread is represented by the (dashed grey line, ×). These scores are averaged over all training stations (a, c) or test stations (b, d), for August in 2020.

For wind components, the spread is consistent with the RMSE of ensemble mean (Fig. 9). This could be attributed to the estimation of the lowest model layer through divergence constraint after wind correction (3DRF), thereby avoiding dispersion attenuation caused by 2D interpolation. The ensemble nowcasting for wind component shows a reliable spread that can be effectively transmitted without causing unusual increases in error. However, when the lead time exceeds 3 hours, the forecast is entirely represented by the first guess. Therefore, using an ensemble of first guesses is a promising approach to improve the uncertainty estimation of wind speed. Additionally, this may enhance the ability of the inflation factor to more comprehensively represent the interpolation uncertainty.

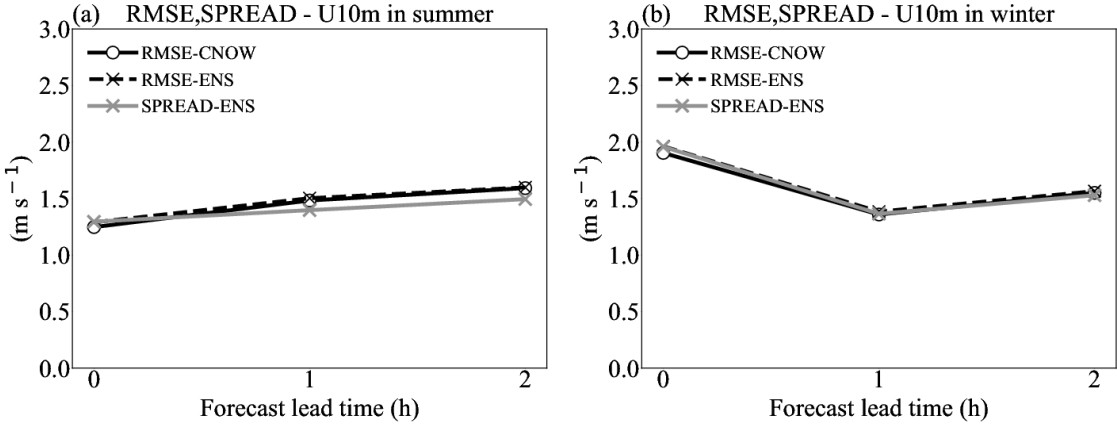

**Fig. 9. RMSE of deterministic nowcasting (black, ○), ensemble mean (black, ✕) and ensemble spread (grey, ✕) for U10m in both August 2020 (a) and February 2021 (b), averaged over all the test stations.**

### 4.2.2 Probability scores

The reliability of T2m is evaluated in Figure 10 in terms of the Talagrand diagram and reliability diagram, valid at lead time +6 hours. The verification reference is the observation at test stations. The rank histogram displays a slight "L-shape", indicating a warm bias in the ensemble nowcasting. One of the reasons is that the persistence of NWP will transmission in the extrapolation. At each time step, the NWP forecast adjust by subtracting the previous step to represent the predictive trend of variables, thereby converting the cold bias into a warm bias in the opposite site. In the reliability diagram, the thresholds are the quartile of the ranked temperature both in summer and winter. At each threshold, the ensemble probabilities closely align with the observed frequency. The high reliability and resolution of T2m ensemble nowcasting at a lead time of 6 hours highlight the effectiveness of the proposed approach in quantifying the nowcast uncertainty.

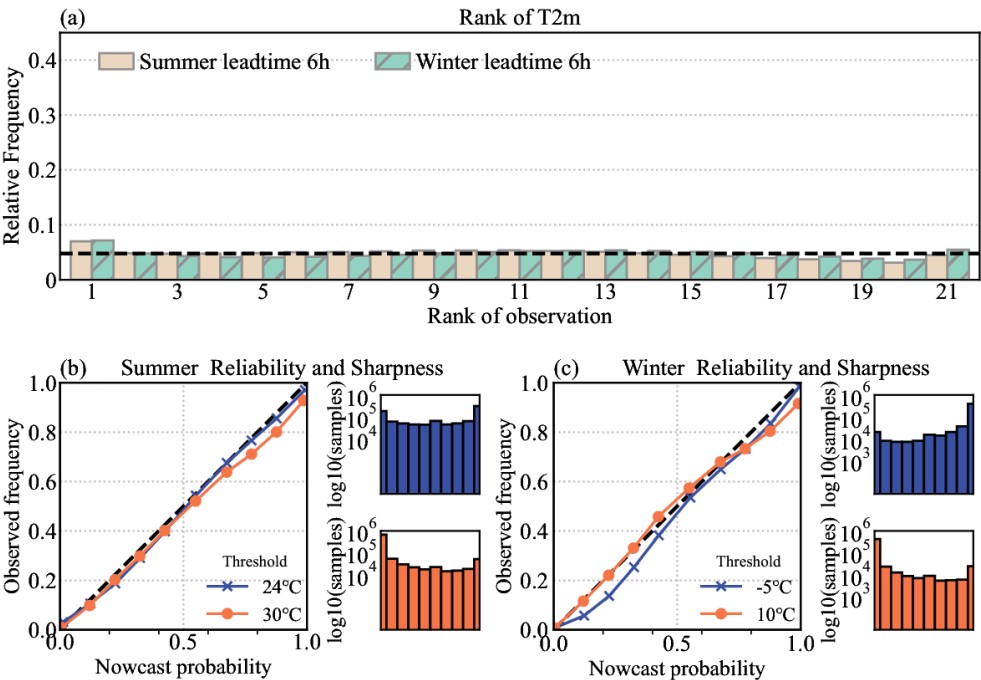

**Fig. 10. Talagrand (a) and reliability diagrams (b, c) for the ensemble nowcasting at lead time of 6 hour of T2m (unit: °C), verified over all the test stations.**

U10m and V10m have similar characteristic in the verification results. The verification of U10m shows that ensemble nowcasting performs with better reliability and resolution in summer compared to winter (Fig. 11). The evident bias for summer U10m, as depicted in the reliability curve (Fig. 11b), suggests that ensemble struggles to capture wind direction uncertainty during periods of significant wind variability. This limitation arises from the lack of appropriate observational perturbations. In the calibration of wind components, vertical wind is used to calculate divergence to constraint the horizontal

wind. Hence, the interpolation of wind differs from that of temperature. The divergence constrains causes the first guess error to incorporate additional information to calibrate the first guess. For this reason, it is difficult to fully understand the impact of the perturbation in observation. Consequently, the reliability of wind components is lower than that of temperature. Further research should address these difficulties by account for the impact of divergence constraint.

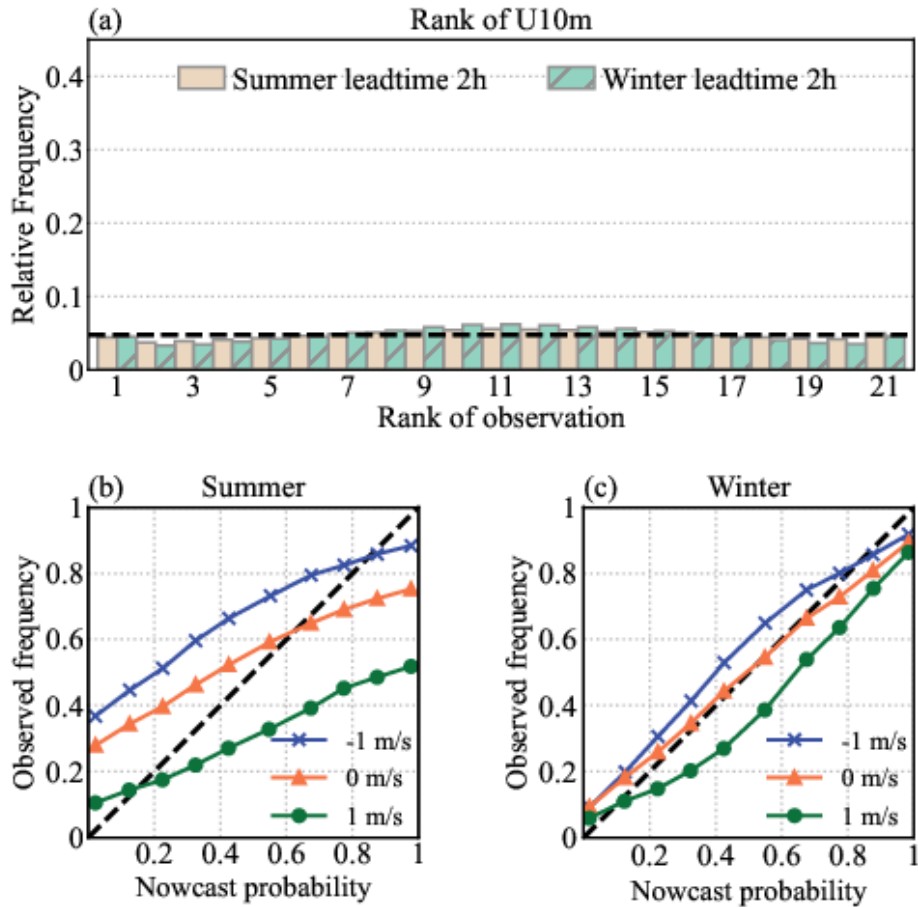

Fig. 11. The same as Fig. 10 but for U10m (unit: m s$^{-1}$).

## 5 Conclusion and discussion

This study proposes an efficient perturbation method to quantify the uncertainty of near–surface atmosphere analysis. Gaussian–distributed noise, generated based on the error characteristic, simulates the propagation of the uncertainty in the analysis computation. An inflation factor is computed to simulate the attenuation of perturbation dispersion during the interpolation process. The ensemble analysis offers a robust estimation of surface uncertainty at the initial time of nowcasting, aligning with the error increment observed in nowcasting extrapolation.

Adding noise to both observations and first guess reflects the dispersion of the analysis error, which reflects the analysis uncertainty, especially in the areas without stations. For temperature, the spatial uncertainty caused by terrain can be addressed by incorporating the perturbed field with terrain information during interpolation. For humidity, its intrinsic correlation with

temperature affects the estimation of the error uncertainty. For wind components, the divergence constraint does not account for variations in perturbation information, resulting in increased RMSE. Addressing this issue could be a focus of future improvements.

The ensemble analysis verification for all variables demonstrates reliable representation of analysis uncertainty. However, wind components suffer the influence of divergence constraint. Relative humidity is influenced by variable conversion processes, leading to an under–dispersive ensemble.

The flat Talagrand diagrams illustrate that ensembles effectively estimate the probable range of true values. Introducing perturbations in analysis computation effectively quantifies the uncertainty of near–surface variables in both magnitude and spatial distribution. A limitation of the current method is that the inflation factor cannot represent the complexity of interpolation uncertainty. A possible improvement would be to see if there are some predictors which could help inflation factors to extrapolate perturbation information to the areas where there are truly no stations.

The ensemble analysis provides a reliable presentation of uncertainty at the initial time of nowcasting. Not only do the errors of ensemble nowcasting match the errors of deterministic nowcasting, but also the growth of ensemble spread also aligns with the error growth trend in nowcasting extrapolation. Since this method does not account for the uncertainty derived by NWP systematic errors and the relativity of different variables, the ensemble nowcasting is slightly under-dispersive.

The perturbed first guess errors, along with the spread and error of the ensemble, are associated with the analysis error observed at test stations. The perturbation method in this study addresses the challenge of accurately representing the uncertainty of near-surface deterministic analysis. This method enhances the estimation of near–surface analysis uncertainty for both nowcasting applications and ensemble nowcasting development. Further improvements could involve considering the uncertainty in estimating the first guess error of multi-source NWP to obtain a more comprehensive spatial uncertainty representation.

*Code and data availability.* The information data, example data and the corresponding codes for generating perturbation are archived on Zenodo: https://doi.org/10.5281/zenodo.11243716. (Zhu, 2024). Due to the confidentiality policy, the code and datasets of SIVA that utilized in this study are not in the public domain and cannot be distributed.

**Appendix A: Case of generating temperature perturbation**

Listing A1 presents how to generate perturbation for both observation and NWP. This case is for 2m temperature while specific humidity and wind components have similar process. The input file sta_inf and grid_inf are the example temperature data and the corresponding standard deviation of first guess error. Due to the confidentiality agreement, these files only include the temperature value. The variable inflation in grid_inf is the inflation factor which is used to rescale the perturbed NWP. Since the standard deviation of perturbation that generated by the function numpy.random.normal exists offsets, the factor sc_sta

and sc_grid are used to ensure the scale of perturbation be consistent with the scale of first guess error.

```python
import xarray as xr
import numpy as np
import pandas as pd
import matplotlib.pyplot as plt

member_number = 10                              # Number of ensemble members
nz = 21                                         # Number of vertical levels
nj = 581                                        # Number of grids in Latitudinal direction
ni = 511                                        # Number of grids in Meridian

# Read the information for generating random noise
sta_inf = pd.read_csv('./observation_data.csv')       # Information in station

grid_inf = xr.open_dataset('./grid_information.nc')  # Information in grid
grid_std = grid_inf.error_std_grid.values             # Standard deviation(STD) of first guess error in grid
nwp_data = grid_inf.nwp.values                        # Example of NWP data
inflation = grid_inf.inflation.values                 # Inflation factor

######################################################
# Generating random noise at each station that use to calculate analysis and nowcasting
sta_noise = np.empty([member_number, len(sta_inf)])
for s_num in range(len(sta_inf)):
    gen_noise_grid = np.random.normal(loc=0, scale=sta_inf['error_std'][s_num], size=member_number)
    # Rescale the standard deviation (STD) of noise
    # to be consistent with the STD of first guess error in each station
    sc_sta = np.std(gen_noise_grid) / sta_inf['error_std'][s_num]
    sta_noise[:, s_num] = (gen_noise_grid - np.nanmean(gen_noise_grid)) / sc_sta

# Generating random noise at each grid point
nwp_noise = np.empty([member_number, nz, nj, ni])
for j in range(nj):
    for i in range(ni):
        gen_noise_grid = np.random.normal(loc=0, scale=grid_std[j, i], size=[member_number, nz])
        sc_grid = np.std(gen_noise_grid) / grid_std[j, i]
        nwp_noise[:, :, j, i] = (gen_noise_grid - np.nanmean(gen_noise_grid)) / sc_grid

# Perturbing NWP
per_nwp = np.empty([member_number, nz, nj, ni])
for x in range(member_number):
    per_nwp[x, :, :, :] = nwp_data + nwp_noise[x, :, :, :]
per_mean = np.nanmean(per_nwp, axis=0)

for x in range(1, member_number + 1):
    # Perturbing observation
    sta_inf['tt'] += sta_noise[x - 1]

    # Inflating the perturbed NWP
    new_nwp = (np.einsum('kij,ij->kij', (per_nwp[x - 1] - per_mean), inflation)) + per_mean

    plt.imshow(new_nwp[0], cmap='jet', origin='lower')
    plt.show()
```

**Listing A1. Process of generating random perturbation and rescaling the perturbed NWP by inflation factor.**

405

*Author contributions.* Yanwei Zhu and Yong Wang equally contributed to this work. Yong Wang and Aitor Atencia proposed the method; Yanwei Zhu applied the method and perform the experiments; Yanwei Zhu wrote the manuscript draft and all authors reviewed and edited the manuscript.

*Competing interests.* The authors declare that they have no conflict of interest.

*Acknowledgements.* We are grateful to Huafeng Meteorological Media Group and the HuaFeng Research Lab for Weather Science and Applications, Nanjing University of Information Science & Technology for their support and technical assistance in this research. The authors express gratitude for funding the three projects on "HUAFENG Forecast Applications" with Grant No.CY-J2020007, CY-J2021002 and CY-2022ZDA01.

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
