# Peer review of "Quantifying the analysis uncertainty for nowcasting application"

_EGUsphere, 2024_

## Author Comment (AC1)

**General statement**

We would like to thank the editor for coordinating the review of our work and the peer-reviewers for their valuable comments on our study. In the following, we will address the referees' comments and present our plans and ideas for revising the manuscript. For clarity, our responses are highlighted in red.

**Referee comment #1**
In this manuscript, the authors claimed that they quantify uncertainty in nowcasting. However, I think the quality of this manuscript does not reach the level of a publishable work for a few key reasons:

The author misunderstood a few fundamental concepts in forecast post-processing. I have not seen the term 'analysis uncertainty' before, and it is very hard to know what it exactly refers to. If you are talking about uncertainties in weather forecasts, you should point that out, rather than using terms like: ' ensemble analysis ', 'analysis uncertainty', 'SIVA uncertainty', etc. In results, the authors tried to compare the ensemble spread with RMSE (e.g., line 203), giving me the impression that they do not really understand the basic statistics of weather forecasts.

**Reply 1:**
Thanks for the comments and we will give more explanations of the used terms in the manuscript. The term "analysis" as discussed in this work refers to "meteorological observations on a grid mesh", which refers to the representation or mapping of meteorological variables (such as temperature, pressure, wind speed, humidity, etc.) over a specific geographical area on a grid mesh with certain resolution (e.g. 1km×1km). Such analysis contains measurement errors and the errors produce by interpolation. We use the term "analysis uncertainty" to describe the uncertainty arising from such errors. This is crucial for gaining a more comprehensive understanding of the uncertainty in nowcasting. Hence, we propose an approach to estimate the uncertainty represented by those errors in the analysis: generating "ensemble analysis" by introducing appropriate perturbations. In the revision, we will ensure that these terms are explicitly defined and used in the proper context.

One of the most commonly used ensemble verification metrics to assess the reliability of ensemble forecasts is to compare the spread of the ensemble with the root-mean-square-error (RMSE) of the ensemble mean (Fortin et al., 2014). The ensemble spread quantifies the dispersion or variability among the ensemble members, while the RMSE measures the accuracy of the ensemble mean relative to the reference values. This comparison, which is a concept widely used in ensemble forecast verification, reflects whether the ensemble spread appropriately represents the uncertainty.

The revision will be traceable in the manuscript.

**Reference:**

Fortin, V., Abaza, M., Anctil, F., Turcotte, R.: Why should ensemble spread match the RMSE of the ensemble mean? J. Hydrometeorol., 15, 1708–1713. https://doi.org/10.1175/JHM-D-14-0008.1, 2014.

The manuscript has been carelessly prepared, making it extremely confusing and hard to understand. The whole manuscript reads like an automatic translation from a foreign language to English, using some software. Lots of grammatical errors and awkward expressions, making it hard to learn what they want to introduce. Please see examples in the detailed comments below.

**Reply 2:**

Thanks for the comment. We will carefully go through the whole paper and check the grammar issue. The revision will be traceable in the manuscript.

Results did not show much improvements, in the resultant ensemble forecasts relative to the original forecasts. I could not figure out the necessity of this work. In addition, many findings presented are based on speculation, rather than based on solid data analysis.

**Reply 3:**

Thanks for the comments. We understand the concern regarding the improvement in ensemble forecasts compared to the original forecasts. The primary objective of this work is not necessarily to show dramatic improvements in forecast scores, but rather to quantify the uncertainty in the analysis using a perturbation approach. We introduce Gaussian perturbations with zero mean into the deterministic analysis to simulate uncertainty. In this context, the scores (BIAS and RMSE) of the ensemble mean should ideally remain consistent with those of the deterministic reference.

Regarding the concern about the necessity of this work, we believe that quantifying uncertainty in the analysis has significant value. This is particularly important for improving the reliability and accuracy of nowcasts in practical applications and addressing the uncertainty in nowcasting. While the method may not lead to major score improvements in this initial evaluation, it provides a framework for understanding the uncertainty inherent in the analysis phase. The key novelty of this work lies in its approach to quantify uncertainty in the analysis and then estimating the uncertainty of nowcasting, rather than focusing solely on forecast score improvements.

We also acknowledge this comment about the speculative nature of some findings. We would like to clarify that all the results presented in this work are based on solid data analysis. For example, the Gaussian perturbations are generated based on the statistical errors in the analysis. We carefully assessed the consistency between the BIAS and RMSE of the ensemble mean and the ensemble spread to ensure that the perturbations do not introduce additional biases, while maintaining an accurate representation of uncertainty.

**Detailed comments:**

1. Line 21, what is the trend of NWP? You should spell out the full name of NWP, when using it for the first time.

Thanks for the comments and sorry for the confusion. We will explain more about this term in the manuscript.

2. Line 37, what does 'the analysis' refer to?

Thanks for the question and sorry for the confusion. The analysis here means "meteorological observations on a grid mesh". We will rephrase the sentence i.e. "As a result, … by those errors in analysis is one of the major challenges …."

3.  Line 42, the 'impact' on what?

We will rephrase the sentence i.e. "Most studies focus on addressing the uncertainty in nowcasting, while only a few have explored the impact of analysis errors on the uncertainty in nowcasting."

4.  Line 43, what is the analysis error

Sorry for the confusion. As described in Reply 1, the analysis is the representation of current atmosphere, calculated by calibrating the first guess using surface observations. We will rephrase the sentence i.e. "… the uncertainty represented by the errors in analysis could have a positive impact …."

5.  Line 52, error produced by interpolation?

Sorry for the confusion. We will revise it i.e. "… into other errors in analysis, such as errors produced by interpolation."

6.  Line 56, a very awkward way of introducing NWP

Thanks for the comments. We will rephrase the sentence i.e. "The first guess used in INCA is the numerical weather prediction (NWP) provided by the Austrian operational version of the Aire Limitée Adaptation dynamique Développement InterNational (ALADIN) limited-area model described by Wang et al. (2006)."

7.  Line 69, to as?

Thanks for the hint. We will revise it, i.e. "The NWP output of China Meteorological Administration Mesoscale model (CMA-MESO) provides a deterministic first guess for SIVA to describes the spatial characteristic (Shen et al., 2020)."

8.  Line 74,  analyses are

Thanks for the hint. We will revise it.

9.  Line 77, for which months?

Sorry for the unclear statement. The periods of our study are August 2020 and February 2021, which are referred to as summer and winter, respectively, in the following text. We will add more details about the dataset description in Section2 of the manuscript.

10. Line 84, how can you calculate analysis

Sorry for the confusion. We will rephrase this description i.e. "The analysis starts with a first guess, which is an NWP short-range forecast output of CMA-MESO. This first guess is then calibrated based on its errors relative to the observations, which are the ground true values provided by automatic weather stations. Topographic parameters are used to map the height of CMA-MESO model levels to the truth altitude of the station location."

11. Line 110, 'Selected…..' this is not a complete sentence

Thanks for the hint, we will revise the sentence i.e. "To assess the effectiveness of ensemble analysis in representing uncertainty, 151 stations were randomly selected as the test set, while the remaining 1519 stations were used for the SIVA computation to generate the ensemble."

12. Line 115, no north arrow, no scale bar, no location information of the study area

Thanks for the comments. We will add north arrow, scale bar and location information in figure1.

13. Line 131, no clear what is 'valley, floor and surface'

Thanks for the hint. We will rephrase it i.e. "…. it implicates the downward (upward) shift that constrains model height to the true altitude of the station location."

14. Line 154, no indentation

Thanks for pointing it out. We will revise it.

15. Line 166, there is no red line in the above figure

Thanks for the hint. We will revise it in this figure.

16. Line 172, why capitalize the word Analysis

Thanks for pointing it out. We will revise it i.e. "Verification of ensemble analysis."

17. Line 183, which summer month and which winter month?

Thanks for the question and sorry for the confusion. We will rephrase it i.e. "…for summer (August 2020) and winter (February 2021)."

---

## Author Comment (AC2)

**General statement**

We would like to thank the editor for coordinating the review of our work and the peer-reviewers for their valuable comments on our study. In the following, we will address the referees' comments and present our plans and ideas for revising the manuscript. For clarity, our responses are highlighted in red.

**Referee comment #2**

This work offers a relevant contribution in the field of meteorological nowcasting, presenting a method to quantify the uncertainty in high resolution analysis at surface level by using a perturbation approach. Nowcasting accuracy is crucial for short term weather events forecast, so this work offers a good contribution because the ideas of using perturbations combined with an inflation factor allows a detailed description of uncertainties derived from the differences between observation and initial estimates. The method has been verified in a specific region through several crucial variables, confirming the robustness of the proposed approach. In my opinion, the manuscript can be considered for publication, but a general revision is needed, in order to address the following comments.

The introduction of the inflation factor represents an important improvement; however, it could be not sufficient to represent the complexity of the interpolation uncertainty. For this reason, I recommend the authors to better clarify the limitation of the proposed approach and to describe the possible potential future improvements.

**Reply 1:**

Thanks for the comments. Before describe the limitation, we will start with "the observation perturbation in this study is …" to describe the generation of perturbation, and then use the comparison of RMSE and spread (the Fig. 2 in manuscript) to explain why we proposed using inflation factor to amplify the spread. This description will be traceable in Section 4. Then, we will explain more about the limitation of the proposed approach in Section4.1.2. The inflation factor is calculated based on the test stations and is then extrapolated to the grid mesh by interpolation, in order to amplify the variance of first guess in the areas without station information. However, this interpolation process is limited by the locations of the test stations, meaning that areas outside these stations can only receive partial information. For this reason, it does not account for the uncertainty of the stations that are not used in the computation of analysis, nor in the computation of the inflation factor (the Fig. 5 in manuscript). Therefore, the limited impact of the inflation factor on the entire grid mesh is a key limitation of the current method. A possible improvement would be to see if there are some predictors which could help inflation factors to extrapolate this information to the outside stations site (not used in the computation of analysis nor in the computation of the inflation factor).

The revision will be traceable in the manuscript.

The method works well with temperature and humidity, but has some difficulties with wind components. The authors claim that it is due to the lack of appropriate observational perturbations,

however I recommend to add a more convincing explanation of this limitation.

**Reply 2:**

 Thanks for the comments. We will add the explanation in the Section4. One reason for the difficulties with wind components is that no perturbation is introduced to the observations. In the calibration of wind components, vertical wind is used to calculate divergence to constraint the horizontal wind. Hence, the interpolation of wind differs from that of temperature. The divergence constrains causes the first guess error to incorporate additional information to calibrate the first guess. For this reason, it is difficult to fully understand the impact of the perturbation in observation. As a result, the reliability of wind components is no as high as that of temperature. Hence, it is necessary to account for the impact of divergence constraint in future research and address these difficulties.

The English style is sufficiently accurate, but it is necessary to improve the readability and clarity. Some complex phrases must be simplified, in order to enhance the readability. There are some grammatical errors; in the following I have reported some errors and imperfections, but there are many others scattered over the text, so a general revision is required.

**Reply 3:**

 Thanks for the comment. We will carefully go through the whole paper and check the grammar issue. The revision will be traceable in the manuscript.

**Detailed comments:**

1. L 29 Avoid repeating the word "nowcasting" twice on the same line.

Thanks for pointing it out. We will rephrase it i.e. "However, due to the chaotic nature of the atmosphere, errors in data and the imperfect numerical models, nowcasting involves uncertainties."

2. L 29 Change "nowcasting is with uncertainties" with "nowcasting involves uncertainties".

Thanks for the comment. We will change this sentence i.e. "… models, nowcasting involves uncertainties." The revision is in line 31.

3. L 36 Avoid repeating the word "uncertainty" twice on the same line.

Thanks for pointing it out. We will rephrase it i.e. "The analysis contains uncertainty, which has a significant impact on nowcasting, due to both measurement errors from instruments and errors produced during the computation."

4. L 51-52 Something is wrong in the English syntax, this sentence is not clear.

Thanks for pointing it out. We will rephrase it i.e. "Horányi et al. (2011) and Bellus et al. (2016, 2019) proved that perturbation can simulate the observation error in analysis. However, they did

not delve into other errors in analysis, such as errors produced by interpolation."

5.  L 54 Change "consider" with "considering".

Thanks for the comments. We will revise it in this sentence.

6.  L 65 (NWP) - I do not understand which NWP are you talking about.

Thanks for pointing it out and sorry for the confusion. We will rephrase this sentence i.e. "The first guess used in INCA is the numerical weather prediction (NWP) provided by the Austrian operational version of the Aire Limitée Adaptation dynamique Développement InterNational (ALADIN) limited-area model described by Wang et al. (2006)" The revision is in line 65-67.

7.  L 66 Haiden et al., 2010, 2011 were already cited at line 64, please remove here.

Thanks for the hint and we will remove this citation in line 66.

8.  L 72-78 This paragraph is confusing and hard to be read. I suggest to remove it and to replace it with a description of the importance of this approach (i.e. strengthens), while this technical description could be moved elsewhere.

Thanks for the comment and sorry for the confusion. We will replace this paragraph as suggested. We will describe the importance of this approach and its potential application prospects. This technical description will be moved to Section2. The revision will be traceable in the manuscript.

9.  L 102-103 "are to match the stations at different altitudes ensure that NWP can be corrected in combination with the topographic parameters at station location". There is something wrong in the English style.

Thanks for pointing it out. We will rephrase it, i.e. "In addition, 21 vertical levels corresponding to various altitudes, such as 0 m, 200 m, and up to 4000 m above the ground, are used to match stations at different elevations. The wind speed is represented in 32 vertical levels with intervals of 125 m. This approach ensures that the first guess can be calibrated by incorporating the topographic parameters at the station location."

10. L 119-121 Avoid using the word "ensemble" five times in the same sentence.

Thanks for the comment. We will rephrase it, i.e. "This work proposes a perturbation method to accurately quantify the uncertainty represented by the errors in analysis. …. The ensemble nowcasting starts at each hour and extrapolates up to a lead time of 2 hours". The revision will be traceable in the manuscript.

11. L 121 "is covered". Do you probably mean "covers"? Otherwise there is a syntax error.

Thanks a lot for pointing it out. We will rephrase this sentence i.e. "… nowcasting covers the test stations shown in Figure 1."

12. L 296 change "for nowcasting at initial time" with "at the initial time of nowcasting".

Thanks for the comment. We will revise it: "… uncertainty at the initial time of nowcasting."

---

## Author Response (AR1)

**General statement**

We would like to thank the editor for coordinating the review of our work and the peer-reviewers for their valuable comments on our study. In the following, we will address the referees' comments and present our plans and ideas for revising the manuscript. For clarity, our responses are highlighted in blue.

**Referee comment #1**

In this manuscript, the authors claimed that they quantify uncertainty in nowcasting. However, I think the quality of this manuscript does not reach the level of a publishable work for a few key reasons:

The author misunderstood a few fundamental concepts in forecast post-processing. I have not seen the term 'analysis uncertainty' before, and it is very hard to know what it exactly refers to. If you are talking about uncertainties in weather forecasts, you should point that out, rather than using terms like: ' ensemble analysis ', 'analysis uncertainty', 'SIVA uncertainty', etc. In results, the authors tried to compare the ensemble spread with RMSE (e.g., line 203), giving me the impression that they do not really understand the basic statistics of weather forecasts.

**Reply 1:**

Thanks for the comments and we will give more explanations of the used terms in the manuscript. The term "analysis" as discussed in this work refers to "meteorological observations on a grid mesh", which refers to the representation or mapping of meteorological variables (such as temperature, pressure, wind speed, humidity, etc.) over a specific geographical area on a grid mesh with certain resolution (e.g. 1km×1km). Such analysis contains measurement errors and the errors produce by interpolation. We use the term "analysis uncertainty" to describe the uncertainty arising from such errors. This is crucial for gaining a more comprehensive understanding of the uncertainty in nowcasting. Hence, we propose an approach to estimate the uncertainty represented by those errors in the analysis: generating "ensemble analysis" by introducing appropriate perturbations. In the revision, we will ensure that these terms are explicitly defined and used in the proper context.

One of the most commonly used ensemble verification metrics to assess the reliability of ensemble forecasts is to compare the spread of the ensemble with the root-mean-square-error (RMSE) of the ensemble mean (Fortin et al., 2014). The ensemble spread quantifies the dispersion or variability among the ensemble members, while the RMSE measures the accuracy of the ensemble mean relative to the reference values. This comparison, which is a concept widely used in ensemble forecast verification, reflects whether the ensemble spread appropriately represents the uncertainty.

The revision will be traceable in the manuscript.

**Reference:**

Fortin, V., Abaza, M., Anctil, F., Turcotte, R.: Why should ensemble spread match the RMSE of the ensemble mean? J. Hydrometeorol., 15, 1708–1713. https://doi.org/10.1175/JHM-D-14-0008.1, 2014.

The manuscript has been carelessly prepared, making it extremely confusing and hard to understand. The whole manuscript reads like an automatic translation from a foreign language to English, using some software. Lots of grammatical errors and awkward expressions, making it hard to learn what they want to introduce. Please see examples in the detailed comments below.

**Reply 2:**

Thanks for the comment. We will carefully go through the whole paper and check the grammar issue. The revision will be traceable in the manuscript.

Results did not show much improvements, in the resultant ensemble forecasts relative to the original forecasts. I could not figure out the necessity of this work. In addition, many findings presented are based on speculation, rather than based on solid data analysis.

**Reply 3:**

Thanks for the comments. We understand the concern regarding the improvement in ensemble forecasts compared to the original forecasts. The primary objective of this work is not necessarily to show dramatic improvements in forecast scores, but rather to quantify the uncertainty in the analysis using a perturbation approach. We introduce Gaussian perturbations with zero mean into the deterministic analysis to simulate uncertainty. In this context, the scores (BIAS and RMSE) of the ensemble mean should ideally remain consistent with those of the deterministic reference.

Regarding the concern about the necessity of this work, we believe that quantifying uncertainty in the analysis has significant value. This is particularly important for improving the reliability and accuracy of nowcasts in practical applications and addressing the uncertainty in nowcasting. While the method may not lead to major score improvements in this initial evaluation, it provides a framework for understanding the uncertainty inherent in the analysis phase. The key novelty of this work lies in its approach to quantify uncertainty in the analysis and then estimating the uncertainty of nowcasting, rather than focusing solely on forecast score improvements.

We also acknowledge this comment about the speculative nature of some findings. We would like to clarify that all the results presented in this work are based on solid data analysis. For example, the Gaussian perturbations are generated based on the statistical errors in the analysis. We carefully assessed the consistency of BIAS and RMSE between the deterministic reference and the ensemble mean to ensure that the perturbations do not introduce additional biases, while maintaining an accurate representation of uncertainty.

**Detailed comments:**

1. Line 21, what is the trend of NWP? You should spell out the full name of NWP, when using it for the first time.

Thanks for the comments and sorry for the confusion. We will explain more about this term in the manuscript in Lines 61-62.

2. Line 37, what does 'the analysis' refer to?

Thanks for the question and sorry for the confusion. The analysis here means "meteorological observations on a grid mesh". We will rephrase the sentence i.e. "As a result, … by those errors in analysis is one of the major challenges …." in Lines 36-37.

3.  Line 42, the 'impact' on what?

We will rephrase the sentence i.e. "Most studies focus on addressing the uncertainty in nowcasting, while only a few have explored the impact of analysis errors on the uncertainty in nowcasting." in Lines 40-42.

4.  Line 43, what is the analysis error

Sorry for the confusion. As described in Reply 1, the analysis is the representation of current atmosphere, calculated by calibrating the first guess using surface observations. We will rephrase the sentence i.e. "Wang et al. (2014) and Suklitsch et al. (2015) presented evidence that introducing additional perturbations to estimate the analysis uncertainty can improve the simulation of nowcasting uncertainty." in Lines 41-42.

5.  Line 52, error produced by interpolation?

Sorry for the confusion. We will revise it i.e. "However, neither ALADIN-LAEF nor C-LAEF addresses the impact of other sources of uncertainty, such as those arising from interpolation." in Lines 49-50.

6.  Line 56, a very awkward way of introducing NWP

Thanks for the comments. We will rephrase the sentence i.e. "The first guess in INCA is the numerical weather prediction (NWP) field, which is provided by the Austrian operational version of the ALADIN limited-area model, as described by Wang et al. (2006)." in Lines 61-62.

7.  Line 69, to as?

Thanks for the hint. We will revise it, i.e. "The NWP output of China Meteorological Administration Mesoscale model (CMA-MESO) provides a deterministic first guess, which is used by SIVA to describe the spatial characteristics (Shen et al., 2020)." in Lines 65-67.

8.  Line 74,analyses are

Thanks for the hint. We will revise it.

9.  Line 77, for which months?

Sorry for the unclear statement. The periods of our study are August 2020 and February 2021,

which are referred to as summer and winter, respectively, in the following text. We will add more details about the dataset description in Section2 of the manuscript, in Lines 99-100.

10. Line 84, how can you calculate analysis

Sorry for the confusion. We will rephrase this description i.e. "The analysis starts with a first guess, which is an NWP short-range forecast output of CMA-MESO. … 2-meter (2m) surface temperature, specific humidity, and 10-meter (10m) surface wind speed." in Lines 83-89.

11. Line 110, 'Selected…..' this is not a complete sentence

Thanks for the hint, we will revise the sentence i.e. "To assess the effectiveness of ensemble analysis in representing uncertainty, 151 stations are randomly selected as the test set, while the remaining 1519 stations are the training set and then are used to generate the ensemble analysis and nowcasting." in Lines 110-112.

12. Line 115, no north arrow, no scale bar, no location information of the study area

Thanks for the comments. We will add north arrow, scale bar and location information in figure1.

13. Line 131,no clear what is 'valley, floor and surface'

Thanks for the hint. We will rephrase it i.e. "…. it implicates the downward (upward) shift that constrains model height to the true altitude of the station location." in Line 145.

14. Line 154, no indentation

Thanks for pointing it out. We will revise it in Line 173.

15. Line 166, there is no red line in the above figure

Thanks for the hint. We will revise it in figure3. The revision is in Lines 185-190.

16. Line 172, why capitalize the word Analysis

Thanks for pointing it out. We will revise it i.e. "Verification of ensemble analysis." in Line 193.

17. Line 183, which summer month and which winter month?

Thanks for the question and sorry for the confusion. We will rephrase it i.e. "…for summer (August 2020) and winter (February 2021)." in Line 204.

**Referee comment #2**

This work offers a relevant contribution in the field of meteorological nowcasting, presenting a method to quantify the uncertainty in high resolution analysis at surface level by using a perturbation approach. Nowcasting accuracy is crucial for short term weather events forecast, so this work offers a good contribution because the ideas of using perturbations combined with an inflation factor allows a detailed description of uncertainties derived from the differences between observation and initial estimates. The method has been verified in a specific region through several crucial variables, confirming the robustness of the proposed approach. In my opinion, the manuscript can be considered for publication, but a general revision is needed, in order to address the following comments.

The introduction of the inflation factor represents an important improvement; however, it could be not sufficient to represent the complexity of the interpolation uncertainty. For this reason, I recommend the authors to better clarify the limitation of the proposed approach and to describe the possible potential future improvements.

**Reply 1:**

Thanks for the comments. Before describe the limitation, we will start with "the observation perturbation in this study is …" to describe the generation of perturbation (in Lines 134-141), and then use the comparison of RMSE and spread (the Fig. 2 in manuscript) to explain why we proposed using inflation factor to amplify the spread. This description will be traceable in Section 3 (in Lines 151-162). Then, we will explain more about the limitation of the proposed approach in Section4.1.2 (in Lines 247-260). The inflation factor is calculated based on the test stations and is then extrapolated to the grid mesh by interpolation, in order to amplify the variance of first guess in the areas without station information. However, this interpolation process is limited by the locations of the test stations, meaning that areas outside these stations can only receive partial information. For this reason, it does not account for the uncertainty of the stations that are not used in the computation of analysis, nor in the computation of the inflation factor (the Fig. 5 in manuscript). Therefore, the limited impact of the inflation factor on the entire grid mesh is a key limitation of the current method. A possible improvement would be to see if there are some predictors which could help inflation factors to extrapolate this information to the outside stations site (not used in the computation of analysis nor in the computation of the inflation factor).

The revision will be traceable in the manuscript.

The method works well with temperature and humidity, but has some difficulties with wind components. The authors claim that it is due to the lack of appropriate observational perturbations, however I recommend to add a more convincing explanation of this limitation.

**Reply 2:**

Thanks for the comments. We will add more explanation in Lines 348-353. One reason for the difficulties with wind components is that no perturbation is introduced to the observations. In the calibration of wind components, vertical wind is used to calculate divergence to constraint the horizontal wind. Hence, the interpolation of wind differs from that of temperature. The divergence constrains causes the first guess error to incorporate additional information to calibrate the first

guess. For this reason, it is difficult to fully understand the impact of the perturbation in observation. As a result, the reliability of wind components is no as high as that of temperature. Hence, it is necessary to account for the impact of divergence constraint in future research and address these difficulties.

The English style is sufficiently accurate, but it is necessary to improve the readability and clarity. Some complex phrases must be simplified, in order to enhance the readability. There are some grammatical errors; in the following I have reported some errors and imperfections, but there are many others scattered over the text, so a general revision is required.

**Reply 3:**

Thanks for the comment. We will carefully go through the whole paper and check the grammar issue. The revision will be traceable in the manuscript.

**Detailed comments:**

1. L 29 Avoid repeating the word "nowcasting" twice on the same line.

Thanks for pointing it out. We will rephrase it i.e. "However, due to the chaotic nature of the atmosphere, errors in data and the imperfect numerical models, nowcasting involves uncertainties." in Lines 28-29.

2. L 29 Change "nowcasting is with uncertainties" with "nowcasting involves uncertainties".

Thanks for the comment. We will change this sentence i.e. "… models, nowcasting involves uncertainties." The revision is in Line 29.

3. L 36 Avoid repeating the word "uncertainty" twice on the same line.

Thanks for pointing it out. We will rephrase it i.e. "The weather analysis contains uncertainty, which significantly impacts nowcasting due to both measurement errors and computational errors." In Lines 34-35.

4. L 51-52 Something is wrong in the English syntax, this sentence is not clear.

Thanks for pointing it out. We will rephrase it i.e. "Horányi et al. (2011) and Bellus et al. (2016, 2019) demonstrated that appropriate perturbations can simulate observation uncertainty in analysis." In Lines 43-44.

5. L 54 Change "consider" with "considering".

Thanks for the comments. We will revise it in Line 50.

6. L 65 (NWP) - I do not understand which NWP are you talking about.

Thanks for pointing it out and sorry for the confusion. We will rephrase this sentence i.e. "The first guess in INCA is the numerical weather prediction (NWP) field, which is provided by the Austrian operational version of the ALADIN limited-area model, as described by Wang et al. (2006)." The revision is in Lines 61-62.

7. L 66 Haiden et al., 2010, 2011 were already cited at line 64, please remove here.

Thanks for the hint and we will remove this citation.

8. L 72-78 This paragraph is confusing and hard to be read. I suggest to remove it and to replace it with a description of the importance of this approach (i.e. strengthens), while this technical description could be moved elsewhere.

Thanks for the comment and sorry for the confusion. We will replace this paragraph as suggested. We will describe the importance of this approach and its potential application prospects in Lines 70-77. This technical description will be moved to Section2. The revision will be traceable in the manuscript.

9. L 102-103 "are to match the stations at different altitudes ensure that NWP can be corrected in combination with the topographic parameters at station location". There is something wrong in the English style.

Thanks for pointing it out. We will rephrase it, i.e. "In addition, 21 vertical levels corresponding to various altitudes, such as 0 m, 200 m, and up to 4000 m above the ground, are used to match stations at different elevations. The wind speed is represented in 32 vertical levels with intervals of 125 m. This approach ensures that the first guess is calibrated using topographic parameters and observation at station location." in Lines 101-104.

10. L 119-121 Avoid using the word "ensemble" five times in the same sentence.

Thanks for the comment. We will rephrase it, i.e. "This work proposes a perturbation method to accurately quantify the uncertainty represented by the errors in analysis." The revision will be traceable in the manuscript in Line 120.

11. L 121 "is covered". Do you probably mean "covers"? Otherwise there is a syntax error.

Thanks a lot for pointing it out. We will rephrase this sentence i.e. "… nowcasting covers the test stations shown in Fig. 1." in Line 126

12. L 296 change "for nowcasting at initial time" with "at the initial time of nowcasting".

Thanks for the comment. We will revise it: "… uncertainty at the initial time of nowcasting." in Line 361.

---

## Author Response (AR2)

**General statement**

We would like to thank the editor for coordinating the review of our work and the peer-reviewers for their valuable comments on our study. In the following, we will address the referees' comments and present our plans and ideas for revising the manuscript. For clarity, our responses are highlighted in blue.

**Referee #2**

1. L 12-13: Change "This error not only reflects the spatial characteristics of the difference between first guess and observation but also dominates the primary uncertainty in analysis errors" with "This error reflects the spatial characteristics of the difference between the first guess and observations and dominates the primary sources of analysis uncertainty."

**Reply 1:**

Thanks for the comments. We have revised the sentence in Line 12-13 as suggested.

2. L 53-55: Change: "The observations provide an estimate of the true atmospheric values, while the three-dimensional first guess provides a complete spatial structure within the region of interest." with "Observations estimate true atmospheric values, while the three-dimensional first guess offers a comprehensive spatial structure for the region of interest."

**Reply 2:**

Thanks for the comments. We have revised the sentence in Line 53-55 as suggested.

3. L 214-215: Change: "The primary objective of this work is not to show dramatic improvements in verification scores, but to focus on the quantification of uncertainty in analysis" with "This work primarily focuses on quantifying analysis uncertainty rather than achieving dramatic improvements in verification scores."

**Reply 3:**

Thanks for the comments. We have revised the sentence in Line 214-215 as suggested.